# Synthesis of Amino-*gem*-Bisphosphonate Derivatives and Their Application as Synthons for the Preparation of Biorelevant Compounds

**DOI:** 10.3390/ph18071063

**Published:** 2025-07-18

**Authors:** Mario Ordoñez, Rubén Oswaldo Argüello Velasco

**Affiliations:** 1Centro de Investigaciones Químicas-IICBA, Universidad Autónoma del Estado de Morelos, Av. Universidad 1001, Cuernavaca 62209, Morelos, Mexico; 2Facultad de Ciencias Químicas e Ingeniería, Universidad Autónoma del Estado de Morelos, Av. Universidad 1001, Cuernavaca 62209, Morelos, Mexico

**Keywords:** synthesis of amino-*gem*-bisphosphonates, synthesis of amino-*gem*-bisphosphonic acids, biological activity of amino-*gem*-bisphosphonates

## Abstract

In recent years, amino-*gem*-bisphosphonic acids and their esters have been considered a family of compounds of great chemical and pharmacological interest due to their important biological properties and their value as key synthons in the synthesis of more complex molecules with biological interest. This explains why several research groups are interested in developing new methods for the preparation of these compounds. Therefore, we would like to report here a summary of the synthetic strategies published in the last fifteen years for the synthesis of acyclic and heterocyclic α-, β- and γ-amino-gem-bisphosphonates, as well as their application in the preparation of selected compounds of chemical and pharmacological interest. This information can be of general knowledge to researchers working in this area, as it provides the starting point for new methods and applications of these compounds.

## 1. Introduction

In recent decades, organophosphorus compounds have been the focus of attention of several research groups, due to the chemical and pharmacological importance of this type of compound. For example, α-aminophosphonic, α-amino-*H*-phosphonic and α-amino-*C*-phosphonic are possibly the most important analogs of α-amino acids, in which the planar carboxylic acid group (-CO_2_H) is replaced by a sterically more demanding tetrahedral phosphonic acid group [-P(O)(OH)_2_] or phosphinic acid [-P(O)(OH)R]. These analogs represent an important class of compounds with multiple applications in medicinal and organic chemistry. This is due to the ability of phosphonic and phosphinic groups to mimic the high-energy transition state of peptide bond hydrolysis, enabling them to act as enzyme inhibitors or receptor ligands in pathological conditions associated with amino acid metabolism [1,2,3,4,5]. Due to the importance of these compounds, excellent methods for their preparation have been published in the last decades [6,7,8,9,10,11,12,13,14,15,16,17,18]. Additionally, *gem*-bisphosphonic acids and *gem*-bisphosphonates are analogs of pyrophosphoric acid and pyrophosphate, respectively, in which the hydrolytically labile P-O-P oxygen bridge is replaced by the hydrolysis-resistant P-C-P linkage, making these compounds metabolically more stable. Furthermore, replacing the methylene hydrogen P-CH_2_-P with other substituents such as the amino group gives rise to compounds of high chemical and biological interest such as the α-amino *gem*-bisphosphonic acids and their corresponding phosphonic esters (Figure 1).

The amino-*gem*-bisphosphonates are an important class of compounds and have received great attention in recent years due to their relevant pharmacological applications. For example, Incadronate **1** is an anticancer agent [19,20,21]; **2** is herbicidal [22]; **3** acts as an antiparasitic agent [23]; the α-amino-*gem*-bisphosphonic acid (BP) linked to Bortezomib (BP-Btz)-**4** is an FDA-approved drug for the treatment of patients with multiple myeloma, more effective and with less side effects than Btz [24]; the α-amino-*gem*-bisphosphonic acid **5** (BPAMD) and α-amino-*gem*-bisphosphonic acid **6** (BPAPD) bearing 1,4,7,10-tetraazacyclododecane-1,4,7,10-tetraacetic acid have been used in imaging (^111^In and ^68^Ga) for single-photon emission computed tomography (SPECT) and positron emission tomography (PET) [25,26]; HBED-CC linked to α-amino-*gem*-bisphosphonic acid (^68^Ga) **7** is a potential PET/CT bone imaging agent, and biodistribution, autoradiography and imaging studies clearly demonstrate that the tracer is taken up almost exclusively by the skeletal bone system, with minimal activity accumulation in other organs [27] (Figure 2).

The biological activity of the selected amino-*gem*-bisphosphonates and amino-*gem*-bisphosphonic acids described in this review are summarized in Table 1.

Despite the wide biological application of the amino-*gem*-bisphosphonic acids, its administration is complicated by poor bioavailability and poor gastrointestinal tolerability. Therefore, the synthesis of amino-*gem*-bisphosphonic acids and their esters with better bioavailability, greater biological activity and lower secondary toxicity is a challenge for chemists [64,65,66,67,68]. In this context, this review aims to summarize the strategies reported in the literature for its synthesis in the last 15 years, as well as their biological properties.

## 2. Synthesis of α-Amino-*gem*-Bisphosphonate Derivatives

The synthesis of α-amino-*gem*-bisphosphonate derivatives can be obtained by four routes: (1) three-component reaction involving orthoformates, amines and dialkyl phosphites; (2) phosphonylation of imidates; (3) phosphonylation of amides; (4) phosphonylation of nitriles; (5) phosphonylation of isonitriles; and (6) electrophilic amination of *gem*-bisphosphonates (Figure 1).

### 2.1. Three-Component Reaction of Orthoformates, Amines and Dialkyl Phosphites

One of the most convenient and common methods for the synthesis of α-amino-*gem*-bisphosphonate derivatives is the three-component reaction involving orthoformates, amines and dialkyl phosphites. For example, Rodriguez et al. [29] carried out the preparation of a series of *N*-alkyl α-amino-*gem*-bisphosphonates **8a**–**i** by reacting triethyl orthoformate with alkyl or cycloalkyl amines and diethyl phosphite at 135 °C, obtaining the corresponding *N*-alkyl α-amino-*gem*-bisphosphonates **8a**–**i** in moderate yields, which, by hydrolysis with HCl at 100 °C, gave the *N*-alkyl α-amino-*gem*-bisphosphonic acids **9a**–**i** in good yields, which were tested against *Trypanosoma cruzi* (amastigotes) and *Toxoplasma gondii* (tachyzoites) (Figure 2).

The mechanism of this three-component reaction has been investigated by Krutikov and Kafarski [69,70]. The first step is the condensation reaction of the amine with the orthoformate, in which imine-type intermediates **A** or **B** may be formed, followed by the nucleophilic addition of diethyl phosphite to the C=N bond of the imines, giving the α-aminophosphonates derivatives **C** and **D**, respectively. Then, the elimination of ethanol or amine molecules, gave the iminophosphonate **E**, which, by the addition of another unit of diethyl phosphite, produced the *N*-substituted α-amino-*gem*-bisphosphonates (Figure 3).

Tortorella et al. [32] reported the reaction of triethyl orthoformate with several arylamines and diethyl phosphite at 160 °C, obtaining the *N*-aryl α-amino-*gem*-bisphosphonates **10a**–**v** in 32 to 89% yield, which, by the hydrolysis of diethyl esters mediated by trimethylsilyl bromide (TMSBr) in anhydrous acetonitrile followed by treatment with methanol, gave the corresponding *N*-aryl amino-*gem*-bisphosphonic acids **11a**–**v** in 21 to 98% yield (Figure 4). These compounds were tested against metalloproteinases, such as MMP-2, MMP-8, MMP-9 and MMP-14.

*N*-sulfonamide *gem*-bisphosphonates **12a**–**h** were obtained in 29 to 92% yield by a three-component reaction of triethyl orthoformate with the appropriate aryl sulfonamide and diethyl phosphite at 150 °C. Hydrolysis of **12a**–**h** with 6 N HCl or BBr_3_ produced the *N*-sulfonamide *gem*-bisphosphonic acids **13a**–**h** in 55 to 98% yield (Figure 5). All compounds obtained were tested against metalloproteinases, such as MMP-2, MMP-8, MMP-9 and MMP-14 [33].

Bochno and Berlicki [71] carried out the three-component reaction of triethyl orthoformate with several amines and ethyl(diethoxymethyl)phosphonate at reflux, to obtain the *N*-substituted α-amino-*gem*-(diethoxymethyl)phosphinates **14a**–**h** in 7 to 50% yield. Hydrolysis of **14a**,**b** with HBr/AcOH followed by treatment with NaHCO_3_ afforded the *N*-substituted α-amino-*gem*-bisphosphonic acids **11a** and **11w** in excellent yield (Figure 6).

In a similar way, Kafarski et al. [31] reported the synthesis of *N*-substituted α-amino-*gem*-bisphosphonic acids **11a**, **11l** and **15a** in 30 to 64% yield through a three-component reaction of triethyl orthoformate, aromatic amines and diethyl phosphite followed by hydrolysis of diethyl esters mediated by TMSBr and subsequent treatment with methanol (Figure 7). The *N*-substituted α-amino-*gem*-bisphosphonic acids **11a**, **11l** and **15a** obtained were evaluated as effective inhibitors of *Mycobacterium tuberculosis* glutamine synthetase.

Three-component reaction of triethyl orthoformate, aromatic amines and diethyl phosphite at 125 °C followed by hydrolysis with 6 N HCl afforded the *N*-aryl α-amino-*gem*-bisphosphonic acids **11l** and **16a**–**j** in 9 to 97% yield, against which their potential antiproliferative activity was evaluated using mouse macrophage-like J774E cells (Figure 8) [34].

Three-component reaction between triethyl orthoformate, *N*,*N*-dibenzylamine and diethyl phosphite followed by the cleavage of *N*-Bn bond under hydrogenolysis afforded the α-amino-*gem*-bisphosphonate **17**, which is a key intermediate for the synthesis of 1,2,3,4-tetrahydroisoquinoline-3-phosphonic acid (±)-**17a** [72]. Wu et al. [35] used the α-amino-*gem*-bisphosphonate as a key intermediate for the synthesis of amides **17b** and **17c**, which were tested as a new class of potential bone-targeting reagents for bone tumors, showing a high affinity to hydroxyapatite in vitro. Additionally, α-amino-*gem*-bisphosphonate **17** was transformed into ligand DOTA-Bn-SCN-BP **17d** for ^68^Ga- and ^153^Sm complexes, and tested as radiopharmaceuticals for both imaging skeletal metastases and palliation of pain arising out of it in cancer patients [73]. α-amino-*gem*-bisphosphonate **17** was also used to obtain the biotin derivative **17e** used as a model linker for protein attachment to bone [74] (Figure 9).

The reaction of triethyl orthoformate with 3-amino-1,2,4-triazole and diethyl phosphite under heating, followed by hydrolysis with HCl, produced 1,2,4-triazoly-3-yl-amino-*gem*-bisphosphonic acid **18** accompanied by the production of significant quantities of *N*-ethylated derivative **19**. 1,2,4-triazoly-3-yl-amino-*gem*-bisphosphonic acid **18** showed interesting activity in anti-osteolytic therapy, as a powerful inhibitor of the activity of J774E cells, and is equipotent to the popular drug Zoledronate and exhibits higher activity than the drug Incadronate (Figure 10) [36].

Tsantrizos et al. [37,38] carried out the three-component reaction of triethyl orthoformate with 2-(methylthio)thieno[2,3-*d*]pyrimidin-4-amine and diethyl phosphite in dry toluene at 130 °C, obtaining the tetraethyl ((2-(methylthio)thieno[2,3-*d*]pyrimidin-4-yl)amino)-*gem*-bisphosphonate **20** in 40% yield, used as a key intermediate in the synthesis of *N*-substituted α-amino-*gem*-bisphosphonic acid derivatives **21a**–**g**, which are identified to exhibit toxicity in various cancer cell lines and are particularly toxic to human myeloma cells, blocking prenylation and inducing apoptosis (Figure 11).

The same research group carried out the three-component reaction of triethyl orthoformate with 6-bromothieno[2,3-*d*]pyrimidin-4-amine and diethyl phosphite in toluene at 130 °C, obtaining tetraethyl ((6-bromothieno[2,3-d]pyrimidin-4-yl)amino)-*gem*-bisphosphonate **22** in 75% yield, which was used as a starting material in the Suzuki cross-coupling reactions to obtain several *N*-substituted α-amino-*gem*-bisphosphonic acid derivatives **23a**–**j** in 17 to 71% yield (Figure 12). The compounds obtained were evaluated as inhibitors of human geranylgeranyl pyrophosphate synthase (hGGPPS) [75].

Tsantrizos et al. [39] carried out the reaction of triethyl orthoformate with diethyl phosphite and the amines **25a**,**b** obtained from **24a**,**b**, at 130 °C, to give the α-amino-*gem*-bisphosphonates **26a**,**b** in 50 and 13% yield, respectively, which were used as key intermediates in the Suzuki coupling reaction for the synthesis of the *gem*-bisphosphonates **27a**–**f** (Figure 13). The *gem*-bisphosphonates obtained were evaluated as inhibitors of the human geranylgeranyl pyrophosphate synthase and their evaluation as antitumor efficacy in multiple myeloma, pancreatic ductal adenocarcinoma and colorectal cancer cells.

Abdou et al. [40] synthesized several *N*-heterocyclic α-amino-*gem*-bisphosphonic acids **28a**–**g** in 70 to 74% yield, reacting ethyl orthoformate with the corresponding amine, diethyl phosphite and metallic sodium in toluene followed by hydrolysis of diethyl ester intermediates with 1 N HCl (Figure 14). The compounds obtained were tested as potential anti-inflammatory agents.

Three-component reaction of triethyl orthoformate with the appropriate 2-aminobenzothiazole and diethyl phosphite at 160 °C afforded the *N*-heterocyclic α-amino-*gem*-bisphosphonates **29a**–**d** in 15 to 53% yield, which, by treatment with 2 N HCl, gave the α-amino-*gem*-bisphosphonic acids **30a**–**d** in 26 to 76% yield. Additionally, *N*-heterocyclic α-amino-*gem*-bisphosphonates **29d**, under hydrogenation of the nitro group using Pd/C, produced the *N*-substituted α-amino-*gem*-bisphosphonate **31** in 77% yield, which was used as a key intermediate in the synthesis of the α-amino-*gem*-bisphosphonic acids **32a**–**f**, whose biological activities were evaluated as MMP-13 inhibitors (Figure 15) [41].

Three-component reactions of triethyl orthoformate with several substituted 3-aminopyridines and diethyl phosphite at 110 °C afforded the *N*-substituted α-amino-*gem*-bisphosphonates **33a**–**q**, which, by O-Et bond cleavage with TMSBr in dichloromethane followed by treatment with methanol, gave the corresponding *N*-substituted α-amino-*gem*-bisphosphonic acids **34a**–**q** in 25 to 65% yield, which were evaluated as inhibitors of human farnesyl pyrophosphate synthase (hFPPS) (Figure 16) [42].

The one-pot reaction of triethyl orthoformate with several arylamines and diethyl phosphite at 130 °C gave the corresponding *N*-substituted α-amino-*gem*-bisphosphonates **35a**–**f**. The compounds obtained induce cytotoxicity and apoptosis in human multiple myeloma cell lines and down-regulate the intended intracellular target in these cells (Figure 17) [43].

Brel [76] carried out the three-component reaction of triethyl orthoformate with *N*-allyl-*N*-methylamine and diethyl phosphite at 70 °C to obtain *N*-substituted α-amino-*gem*-bisphosphonate **36** in 72% yield, which, by [3 + 2] cycloaddition with nitrile oxides to the double bond at −40 °C, gave the 4,5-dihydroisoxazoles **37a**–**d** in good yield which may be useful for drug design and fine organic synthesis (Figure 18).

Chmielewska et al. [44,77] carried out a detailed study of the reaction of diamines with triethyl orthoformate and diethyl phosphite followed by hydrolysis with HCl. In the case of *trans*-cyclohexane-1,4-diamine **38** and *trans*-cyclohexane-1,3-diamine **40**, which gave cyclohexane-1,4- and cyclohexane-1,3-di(aminomethylenebisphosphonic) acids *trans*-**39** and *trans*-**41** in 52 and 51% yield, respectively, derived from the reaction of the two amino groups. On the other hand, when (1*R*,2*R*)-cyclohexane-1,3-diamine **42** and (1*S*,2*S*)-cyclopentane-1,2-diamine **44** were used as starting reagents, (1*R*,2*R*)-4-cyclohexane-1-amino-2-aminomethylenebisphosphonic acid (1*R*,2*R*)-**43** and (1*S*,2*S*)-cyclopentane-1-amino-2-aminomethylenebisphosphonic acid (1*S*,2*S*)-**45** were obtained in 19 and 17% yield, respectively, derived from the reaction of the only one amino group (Figure 19). Other diamines were also used in the study. The antiproliferative action of these di(aminomethylenebisphosphonic) acids and 1-amino-2-amino-*gem*-bisphosphonic acids obtained were evaluated against mouse macrophage-like RAW 264.7.

Three-component reaction of triethyl orthoformate with primary and secondary amines and diethyl phosphite under microwave (MW) irradiation in the absence of catalyst and solvent produced the corresponding *N*-substituted and *N*-disubstituted α-amino-*gem*-bisphosphonates **8b**, **8g**, **10a** and **46a**–**g** in 61 to 86% yield (Figure 20) [78].

Additionally, the three-component reaction of triethyl orthoformate with aminoadamantanes **47a**,**b** and **48** and diethyl phosphite under microwave irradiation (400 W, 150 °C) without catalyst and under solvent-free conditions gave the α-amino-*gem*-bisphosphonates **49a**,**b** and **50** containing a biologically active adamantyl fragment. Reaction of α-amino-*gem*-bisphosphonates **49a**,**b** with TMSBr followed by treatment with methanol and propylene oxide at room temperature produced α-amino-*gem*-bisphosphonic acid **51a**,**b** in good yield (Figure 21) [79].

In a similar way, Cirandur et al. [45] reported a simple, effective and green procedure for the synthesis of α-amino-*gem*-bisphosphonates. In this context, the one-pot reaction of triethyl orthoformate with aromatic amines and diethyl or methyl phosphite without catalyst or solvent under microwave irradiation at 600 W afforded the corresponding α-amino-*gem*-bisphosphonates **29a**, **29e**–**g** and **52a**–**f** in 72 to 93% yield. (Figure 22). The compounds obtained were tested in vitro for their antibacterial, antifungal and antioxidant activity. Molecular docking studies were also performed.

Three-component reaction of triethyl orthoformate with appropriate sulfamides and diethyl phosphite under microwave irradiation at 150 °C and 500 W produced the corresponding α-amino-*gem*-bisphosphonates **12a**,**b** and **53a**–**d** in 50 to 65% yield, which were tested in vitro for their anti-inflammatory activity, showing moderate inhibition compared with Diclofenac. Furthermore, to rationalize the observed biological data, several in silico approaches were used to explain the structure and activity (Figure 23) [48].

Reaction of triethyl orthoformate with 9-(5-amino-5-deoxy-2,3-*O*-isopropylidene-β-D-ribofuranosyl)-6-(2,5-dimethylpyrrol-1-yl)purine **54** and diethyl phosphite under microwave irradiation at 125 °C produced the corresponding *N*-substituted α-amino-*gem*-bisphosphonate **55** in 34% yield, which was transformed into 5′-deoxy-5′-*N*-(methylene bisphosphonate)adenosine (tetra sodium salts) **56** and evaluated for CD73 inhibition in a cell-based assay (MDA-MB-231) and towards the purified recombinant protein (Figure 24) [30].

Cirandur et al. [46] carried out a detailed study of the three-component reaction of triethyl orthoformate with arylamines and diethyl phosphite in the presence of various catalysts such as FeCl_3_, AlCl_3_, LaCl_3_, ZnCl_2_, NiCl_2_, CuCl_2_, CuBr_2_, BF_3_^.^SiO_2_, Fe_3_O_4_ and TiO_2_; however, under these conditions no favorable yields of α-amino-*gem*-bisphosphonates were obtained. The best yields were obtained when the reaction of triethyl orthoformate with several aryl, heteroaryl amines and diethyl phosphite was carried out in the presence of CuO nanoparticles (NPs) as catalysts under microwave irradiation at 400 W and solvent-free conditions (various solvents and watts were also evaluated), obtaining the corresponding *N*-substituted α-amino-*gem*-bisphosphonates **29e**, **57a**–**i** in 92 to 96% yield, which exhibited significant antioxidant and considerable antimicrobial activities (Figure 25).

Reddy et al. [47] carried out a detailed study of the three-component reaction of triethyl orthoformate with arylamines and diethyl phosphite in the presence of various catalysts such as FeCl_3_, ZnCl_2_, I_2_, CuCl_2_, AlCl_3_, *p*-TSA, BF_3_^.^SiO_2_, Amberlyst 15 and nano ZnO at 100 °C, finding that nano ZnO was the most effective catalyst. Thus, the three-component reaction of triethyl orthoformate with arylamines and diethyl phosphite in the presence of a catalytic amount of nano ZnO as an environmentally benign and heterogeneous catalyst under solvent-free and microwave irradiation at 400 W produced the corresponding α-amino-*gem*-bisphosphonates **57c**–**e** and **57j**–**p** in excellent yield (Figure 26). The obtained compounds were evaluated in five cancer cell lines, including human breast (MCF7), prostate (DU-145), osteosarcoma (MG-63), fibrosarcoma (HT-1080) and multiple myeloma (RPMI-8226), showing promising cytotoxic activity in the five cell lines. Compound **57j** was two times more active than Adriamycin in all five cancer cell lines.

In order to develop a clean, eco-friendly, environmentally benign and economical sustainable protocol for the synthesis of α-amino-*gem*-bisphosphonates, Cirandur et al. [80] carried out a detailed study of the three-component reaction of triethyl orthoformate with arylamines and diethyl phosphite in the presence of various catalysts such as ZnCl_2_, FeCl_3_, AlCl_3_, NiCl_2_, Ni(acac)_2_, TiO_2_, LaCl_3_, Rh(OAc)_4_, CuCl_2_, CuBr_2_ and rGO-SO_3_H at room temperature to 100 °C, finding that rGO-SO_3_H was the better catalyst. Thus, the one-pot reaction of triethyl orthoformate with several anilines and diethyl phosphite in the presence of sulfonated reduced graphene oxide (rGO-SO_3_H) as a heterogeneous reusable catalyst under microwave irradiation and solvent-free conditions afforded the *N*-substituted-substituted α-amino-*gem*-bisphosphonates **10b**, **10g**, **10l**, **57a** and **58a**–**e** in 85 to 94% yield (Figure 27). The compounds obtained were evaluated for their anticancer activity against human breast cancer in MCF-7 cell lines, and molecular docking studies were also carried out against human estrogen receptor alpha (ERα).

In a similar way, the three-component reaction of triethyl orthoformate with aryl or 2-pyridylamine, diethyl phosphite and sulfated choline ionic liquid (SCIL) as a recyclable catalyst at room temperature, under conventional heating (Method A) or under ultrasonication (Method B), afforded the *N*-substituted α-amino-*gem*-bisphosphonates **10**, **58b**, **58f**–**n** in 75 to 85% yield (Method A). Higher yields (87 to 95%) were obtained when the reaction was carried out under ultrasonication at room temperature (Figure 28) [81].

On the other hand, Prishchenko et al. [82] carried out the reaction of triethyl orthoformate with several 4-aryl or 2-pyridylamines and diethyl phosphite in the presence of ZnCl_2_ at 140 to 150 °C under solvent free conditions, obtaining the *N*-substituted α-amino-*gem*-bisphosphonates **10a**–**c**, **58i**, **58o**,**p**, **59a**–**d** in 42 to 62% yield (Method A). Higher yields (84 to 92%) were obtained when the four-component reaction of triethyl orthoformate, 4-aryl or 2-pyridylamines, diethyl phosphite and diethyl (trimethylsilyl) phosphite was carried out (Method B) (Figure 29).

After several experiments, Kim et al. [83] found that the one-pot reaction of triethyl orthoformate, arylamines and diethyl phosphite in the presence of sulfonated micro-porous hyper-cross-linked 2,2′-biphenol polymer (HCBP-SO_3_H) as a catalyst in a vial sealed at 50 °C gave the corresponding *N*-substituted α-amino-*gem*-bisphosphonates **10h**, **10j**, **10l**, **58b**, **58i**–**l** and **60a**–**m** in 88 to 95% yield (Figure 30).

Cirandur et al. [84] carried out the reaction of triethyl orthoformate, various amines and diethyl phosphite in the presence of amberlyst-15 as the catalyst at room temperature under solvent free conditions to obtain the *N*-substituted α-amino-*gem*-bisphosphonates **10b**, **10e**, **10p**, **29a** and **61a**–**f** in 70 to 95% yields. These compounds exhibited significant antioxidant properties in nitric oxide method inhibitory potency and antimicrobial activity (Figure 31).

Ionic liquids (ILs) have attracted the attention of various researchers to reduce or eliminate the production and use of harmful and toxic chemicals, which has led to the development of much more efficient, improved and environmentally friendly processes and products [85,86,87,88]. For example, Jeong et al. [89], after comparison of several ionic liquids, found that the di-*n*-butyl ammonium ionic liquid (DIBA IL) as a recyclable ionic liquid is a good catalyst in the three-component reaction of triethyl orthoformate, arylamines and diethyl phosphite at 60 °C, producing the corresponding *N*-substituted α-amino-*gem*-bisphosphonates **10b**–**f**, **29a**, **58j**, **58o,p** and **62a**–**g** in 89 to 94% yield (Figure 32).

In a similar way, after a detailed study using several catalysts, it was found that the three-component reaction of triethyl orthoformate with 4-aryl substituted thiazol-2-amines **63a**–**l** and dialkyl or aryl phosphites in the presence of catalytic amounts of Ag nanoparticles (NPs) at 60 °C under solvent free conditions afforded the *N*-substituted α-amino-*gem*-bisphosphonates **64a**–**l** in 85 to 94% yield (Figure 33). Computational docking methods were used to predict how these α-amino-*gem*-bisphosphonates compete against the inhibitor BPH-1330 at the crystal enzyme structure of the 4H3A protein active site and how the substituent influences their binding ability [90].

On the other hand, the reaction of triethyl orthoformate with various α-amino acids, diethyl and dimethyl phosphite in a EtOH/H_2_O mixture at reflux afforded the α-amino acid-substituted α-amino-*gem*-bisphosphonates **65** and **66** in 54 to 77% yield (Method A). The α-amino acid-substituted α-amino-*gem*-bisphosphonates **65** and **66** were obtained in higher yields (72 to 85%) when the reaction was catalyzed by Tween 20 in aqueous medium at 70 °C (Method B) (Figure 34) [91]. The α-amino-*gem*-bisphosphonates obtained showed in vitro antibacterial activity against clinically isolated bacteria *Klebsiella pneumonia*, *Pseudomonas aeruginosa* (Gram+) and *Staphylococcus aureus*, *Bacillus subtilis* (Gram−). Molecular docking studies against the bacterial target enzyme Type IIA topoisomerase were also carried out to establish the protein–ligand interactions.

### 2.2. Phosphonylation of Imidates

Phosphonylation of readily accessible alkyl imidate hydrochlorides is another methodology for the synthesis of tri- and tetrasubstituted α-amino-*gem*-bisphosphonates. The three-step synthesis involves acylation of the imidate hydrochloride, the addition of diethyl phosphite to the *N*-acylimidate and subsequent nucleophilic substitution of the ethoxy group in the 1-ethoxyphosphonate derivative with triphenylphosphonium tetrafluoroborate [92,93]. Under this method, Kuźnik et al. [94,95] carried out the reaction of ethyl imidate hydrochlorides **67a**–**g** with benzyl chloroformate and Hünig’s base, obtaining the ethyl *N*-(benzyloxycarbonyl)phenylacetimidates **68a**–**g**, which were reacted with diethyl phosphite, potassium carbonate and 18-crown-6 in acetonitrile at 70 °C, affording the diethyl 1-(*N*-benzyloxycarnonylamino)-1-ethoxyalkylphosphonates **69a**–**g**. Subsequently, the reaction of **69a**–**g** with triphenylphosphonium tetrafluoroborate followed by treatment with triethyl phosphite, produced the corresponding tetrasubstituted *N*-Cbz-α-amino-*gem*-bisphosphonates **71a**–**f** in 40 to 95% yield via the 1-(*N*-benzyloxycarbonylamino)-1-triphenylphosphonium-methylphosphonate tetrafluoro-borate salts **70a**–**f**. Additionally, the reaction of triphenylphosphonium-methyl-phosphonate tetrafluoroborate salts **70a**–**g** with diethyl phenylphosphonite at 40 °C, gave the tetrasubstituted phosphonyl/phosphinyl derivatives **72a**–**g** in 62 to 95% yield, with 1:1 to 1:1.7 diastereomeric ratio (Figure 35).

Following the same procedure described above, the ethyl formidate hydrochloride was transformed into *N*-Cbz-α-amino-*gem*-bisphosphonate **73**, which, by hydrogenolysis using Pd(OH)_2_/C, afforded α-amino-*gem*-bisphosphonate **17**, used in the synthesis of bisphosphonate/betulin derivatives **74** and **75** and evaluated as a cytotoxic agent (Figure 36) [96].

Reaction of *N*-aryl ethyl imidates **76a**,**b** with diethyl phosphite in the presence of ZnCl_2_ under heating afforded the corresponding *N*-aryl-α-amino-*gem*-bisphosphonates **10a**,**b** in 57 and 62% yield, respectively. Additionally, the three-component reaction of *N*-aryl ethyl imidates **76a**,**b** with bis(trimethylsilyl) phosphite and tris(trimethylsilyl) phosphite catalyzed by ZnCl_2_ under heating, produced the trimethylsilyl *N*-aryl-α-amino-*gem*-bisphosphonates **77a**,**b** in 85 and 83% yield, respectively; which, by treatment with MeOH, gave the *N*-aryl-α-amino-*gem*-bisphosphonic acids **11a**,**b** in 97 and 96% yield, respectively. (Figure 37) [45].

*N*,*O*-Bis(trimethylsilyl)acetamides are also excellent electrophiles towards the attack of nucleophilic reagents such as silylated phosphonites for the synthesis of α-amino-*gem*-bisphosphonic acids. For example, the reaction of the amides **78a**–**i** with trimethylsilyltrifloromethanesulfonate (TMSOTf) in an anhydrous pentane/dichloromethane mixture at room temperature gave the imidates **79a**–**i**, which, by reaction with anhydrous hypophosphorous acid in the presence of ZnI_2_ in THF at 0 °C, produced the corresponding tetrasubstituted α-amino-*gem*-bisphosphinic acid derivatives as disodium salts **80a**–**i** in 35 to 79% yield, through bis(trimethylsilyl)phosphonite and *N*-silylacetamide intermediates (Figure 38) [97].

Petrosyan et. al. [98] carried out the reaction of imidates **81a**,**b** with diethyl phosphite and diethyl (trimethylsilyl) phosphite, obtaining the α-amino-*gem*-bisphosphonates **82a**,**b** in 42 and 34% yield, respectively (Figure 39).

The Arbuzov reaction of triethyl phosphite with *N*-dichloromethylenetrifloroacetamide **83** obtained from the photochemical chlorination of *N*-methyltrifluoroacetamide gave the *N*-trifluoroacetyl α-amino-*gem*-bisphosphonate **84** in 20% yield, derived from the substitution of both chlorine atoms in **83** (Figure 40) [99].

### 2.3. Phosphonylation of Amides

Phosphonylation of amides is another method for the synthesis of α-amino-*gem*-bisphosphonates. For example, the reaction of 2-(*N*-formyl)-aminopyridines **85a**–**h** with tris(trimethylsilyl)phosphine and trimethylsilyltrifloro-methanesulfonate as activating agents gave the corresponding tetra(trimethylsilyl) α-amino-*gem*-bisphosphonates **86a**–**h**, which, without further purification, were treated with MeOH to give the *N*-pyridyl-α-amino-*gem*-bisphosphonic acids **87a**–**h** in 85 to 96% yield (Figure 41) [45].

Cheviet and Peyrottes [100] used *N*-Cbz aziridines **88a**–**g** as the starting material for the preparation of brand *gem*-bisphosphonylaziridine derivatives. Thus, the reaction of diethyl phosphite with LiHMDS followed by the addition *N*-Cbz aziridines **88a**–**g** in THF at r.t. or 80 °C afforded the *gem*-bisphosphonylaziridines **89a**–**g** (abbreviated as AzbisPs) in 0 to 97% yield (Figure 42).

Cativiela et al. [101] reported the reaction of formamide with phosphorus acid and phosphorus trichloride at 70 °C followed by treatment with water, obtaining the α-amino-*gem*-bisphosphonic acid **90** in 53% yield, which was used as a key intermediate in the synthesis of α-amino-*gem*-bisphosphonate **91** incorporating L-leucine (Figure 43).

On the other hand, the reaction of *N*-Boc *N*-(3-aminopropyl)acetamide **92** with phosphorus oxychloride and triethylphosphite at room temperature produced the *N*-substituted α-amino-*gem*-bisphosphonate **93** in 80% yield, which, by treatment with TMSBr in dichloromethane followed by the addition of methanol, gave the corresponding α-amino-*gem*-bisphosphonic acid **94** in 52% yield. Additionally, the reaction of *N*-substituted α-amino-*gem*-bisphosphonate **93** with trifluoroacetic acid afforded the *N*-substituted α-amino-*gem*-bisphosphonate **95** in 94% yield (Figure 44). The compounds obtained were used for the preparation of organometallic complexes and their cytotoxicity was assessed in vitro using various histologically different cell line models [102].

In a similar way, the reaction of *N*-(2-aminoethyl)acetamide hydrochloride **96** with phosphorus oxychloride and triethyl phosphite at room temperature gave the corresponding *N*-substituted α-amino-*gem*-bisphosphonate **97** in 61% yield, which was transformed into acetyl ursolic acid derivative **98**. Additionally, the reaction of acetamide **99** derived from acetyl ursolyl chloride, with phosphorus oxychloride and triethyl phosphite at room temperature, produced also the acetyl ursolic acid derivative **98** in 71% yield (Figure 45) [103].

The reaction of *N*-formyl heterocyclic derivatives **100a**–**f** with tris(trimethylsilyl)-phosphite and TMSOTf as an activating agent in dichloromethane at 90 °C produced the corresponding tetra(trimethylsilyl) α-amino-*gem*-bisphosphonates **102a**–**f** via the highly reactive trimethylsilyl phosphonates intermediates **101a**–**f**. Treatment of tetra(trimethylsilyl) α-amino-*gem*-bisphosphonates **102a**–**f** with methanol gave the corresponding α-amino-*gem*-bisphosphonic acids **103a**–**f** (Figure 46) [104].

In a similar way, the reaction of *N*-formyl amino acid derivatives **104a**–**h** with tris(trimethylsilyl)phosphite and TMSOTf as activating agents in dichloromethane at 20 °C produced the corresponding tetra(trimethylsilyl) α-amino-*gem*-bisphosphonates **105a**–**h**, which, by treatment with methanol, gave the α-amino-*gem*-bisphosphonic acids **106a**–**h** in 88 to 97% yield (Figure 47) [105,106].

Additionally, the reaction of amines **107a**–**n** with formic acid followed by treatment with *N*-tris(trimethylsilyl)phosphite and TMSOTf in dichloromethane at 20 °C and subsequent treatment with methanol afforded the *N*-substituted α-amino-*gem*-bisphosphonic acids **108a**–**n** in excellent yield (Figure 48) [106].

Phosphonylation of both secondary and tertiary amides activated with trifluoromethanesulfonic anhydride (Tf_2_O) is also a mild and general method for the synthesis of *N*-substituted α-amino-*gem*-bisphosphonates. For example, the reaction of secondary and tertiary amides **109a**–**ab** with diethyl phosphite in the presence of Tf_2_O and 2,6-lutidine or 2,6-di-*tert*-butyl-4-methylpyridine (DTBMP) at 0 °C produced the corresponding α-amino-*gem*-bisphosphonates **110a**–**ab** in 57 to 94% yield (Figure 49) [107].

### 2.4. Phosphonylation of Nitriles

In the Ritter reaction, the reactive intermediate nitrilium has been used in the synthesis of a wide variety of compounds of chemical and pharmacological interest [108,109]. For example, the reaction of trimethyl orthoformate and the nitriles **111a**–**z** in the presence of catalytic amounts of Tf_2_O gave the reactive Ritter intermediate nitrilium, which, by the addition of diethyl phosphite at 60 °C, produced the α-amino-*gem*-bisphosphonates **112a**–**z** in 35 to 90% yield (Figure 50) [110].

Kaboudin et al. [111] carried out a detailed study of the reaction of phosphonylation of nitriles with diethyl phosphite in the presence of various catalyst such as FeCl_3_, BiCl_3_, Sc(OTf)_3_, TiO_2_, ZnO and Zn(OAc)_2_; however, under these conditions, no favorable yield of the α-amino-*gem*-bisphosphonates was obtained. The best yields were obtained when the reaction of nitriles **111a**–**d**, **111s,t** and **113a**–**i** with diethyl phosphite and triethylamine was carried out in the presence of catalytic amounts of ZnCl_2_ in dichloromethane at reflux, obtaining the α-amino-*gem*-bisphosphonates **114a**–**o** in 35 to 75% yield (Figure 51).

Islas and García [112] carried out a detailed study of the reaction of diisopropyl phosphite with benzonitrile **111a** in the presence of various catalysts such as, NiCl_2_.6H_2_O, B(CH(Me)(Et))_3_, AlCl_3_, BF_3_^.^OEt_2_ and Et_3_B; however, in any case, α-amino-*gem*-bisphosphonate was produced. Additionally, when the reaction of diisopropyl phosphite was carried out with benzonitrile catalyzed with NiCl_2_^.^6H_2_O at 140 °C, it gave the α-amino-*gem*-bisphosphonate **115** and α-aminophosphonate **116** with a conversion of 98% and 81:12 ratio (Figure 52).

Wiemer et al. [49] carried out the reaction of nitrile **117** with diethyl phosphite, ZnCl_2_ and triethylamine to obtain the corresponding α-amino-*gem*-bisphosphonate **118** in 42% yield, which, by treatment with TMSBr followed by the addition of methanol, gave the α-amino-*gem*-bisphosphonic acid **119** bearing a triazole fragment in 42% yield (Figure 53). This compound exhibited an inhibitory activity against geranylgeranyl diphosphate synthase (GGDPS).

Titanium-mediated double phosphonylation of nitriles is another procedure for the preparation of α-amino-*gem*-bisphosphonates. For example, the reaction of several nitriles **111e**,v**111q**–**s** and **113j**–**m** with diethyl phosphite catalyzed by bis(cyclopentadienyl)titanium dichloride (Cp_2_TiCl_2_) activated Zn powder and propylene oxide in THF at reflux, affording the corresponding α-amino-*gem*-bisphosphonates **114e** and **120a**–**h** in 20 to 94% yield (Figure 54) [113].

The reaction of aldehydes with hydroxylamine hydrochloride in DMSO at 90 °C afforded the corresponding nitrile, which, without further purification or separation, reacted with diethyl phosphite in the presence of ZnCl_2_ and Et_3_N at 70 °C, obtaining the α-amino-*gem*-bisphosphonates **114f**–**j** and **114p** in 47 to 63% yield (Figure 55) [111].

The reaction of nitriles **111a**, **111q** and **113o** with phosphorus acid (H_3_PO_3_) in the presence of phosphorus trichloride (PCl_3_) at 80 °C followed by treatment with water at 80 °C produced the corresponding α-amino-*gem*-bisphosphonic acids **121a**–**c** in good yield (Figure 56) [98].

Ewies et al. [50] reported the synthesis of tetrakisphosphonic acid derivatives **122a**–**g** in 45 to 65% yield by reacting the dinitriles with phosphorus acid and phosphorus trichloride at 50 °C (Figure 57). The obtained compounds were evaluated for their farnesyl pyrophosphate synthase (FPPS) inhibitory activity and anti-osteoclastogenic properties in vitro using the MTT assay and the tartrate-resistant acid phosphatase (TRAP) staining test.

### 2.5. Phosphonylation of Isonitriles

Isonitriles are also starting reagents for the synthesis of α-amino-*gem*-bisphosphonates. For example, the reaction of isonitriles **123a**–**h** with triethyl phosphite in the presence of hydrochloric acid in dichloromethane at −15 °C afforded the corresponding *N*-substituted α-amino-*gem*-bisphosphonates **124a**–**g** in 84 to 98% yield, which, by hydrolysis with 6 M hydrochloric acid at reflux, produced the α-amino-*gem*-bisphosphonic acids **125a**–**h** in 63 to 92% yield (Figure 58). These compounds were evaluated for their antiproliferative effect on MCF-7 human breast cancer cells, J774E mouse macrophages cells and HL-60 human promyelocytic leukemia cells [51,114].

Sheldon et al. [28] carried out the reaction of isonitriles **126a**–**e** with triethyl phosphite and 4 M HCl in dichloromethane at 0 °C, obtaining the *N*-substituted α-amino-*gem*-bisphosphonates **8b**, **8j**, **8k** and **10y**, **10z** in 50 to 77% yield, which, by treatment with TMSBr followed by the addition of methanol, gave the *N*-substituted α-amino-*gem*-bisphosphonic acids **9b**, **9j**, **9k** and **11y**, **11z** in 69 to 97% yield (Figure 59). These compounds provided cytoprotection against cholesterol-dependent cytolysins. In the work, Sheldon reported also the synthesis of α-amino-*gem*-bisphosphonic acids from triethyl orthoformate.

On the other hand, the reaction of 1,4-diisocyanobenzene **127** and 1,5-diisocyanonaphthalene **129** with triethyl phosphite in the presence of hydrochloric acid in dichloromethane at −15 °C, followed by treatment with TMSBr in dichloromethane and the subsequent addition of methanol, produced benzene-1,4-bis[amino methylidene(bisphosphonic)] acid **128** and naphthalene-1,5-bis[amino methylidene(bisphosphonic)] acid **130** (Figure 60). The antiproliferative activity of these α-amino-*gem*-bisphosphonic acids in combination with doxorubicin and cisplatin toward J774E cells (a model of osteoclast precursors in vitro) was evaluated [52,115].

### 2.6. Electrophilic Amination of gem-Bisphosphonates

The electrophilic amination is a powerful approach for the synthesis of useful intermediates such as α-amino aldehydes, α-amino ketones and α-amino acids [116,117,118]. In this context, the electrophilic amination of *gem*-bisphosphonates is also another method for the synthesis of α-amino-*gem*-bisphosphonates. For example, the reaction of tetraallyl methylenebisphosphonate **131** with sodium hydride in DMF followed by the addition of *O*-diphenylphosphinylhydroxylamine (DPPH) gave the α-amino-*gem*-bisphosphonate **132** in 59% yield, which was a key intermediate in the preparation of α-amino-*gem*-bisphosphonate glycopeptide derivative **133**, used as prodrugs in the treatment of osteomyelitis (Figure 61) [119].

## 3. Synthesis of β-Amino-*gem*-Bisphosphonate Derivatives

The synthesis of β-amino-*gem*-bisphosphonate derivatives can be obtained by two routes: (1) Michael-type addition of amines to vinyl *gem*-bisphosphonates and (2) addition of bisphosphonates to imines (Figure 62).

### 3.1. Michael-Type Addition of Amines to Vinyl gem-Bisphosphonates

The Michael-type addition of cycloalkylamines to tetraethyl vinyl *gem*-bisphosphonate **134** at room temperature in CH_2_Cl_2_ afforded the corresponding *N*-cycloalkyl β-amino-*gem*-bisphosphonates **135a**–**c** in excellent yields, which, by hydrolysis with hydrochloric acid, produced the *N*-cycloalkyl β-amino-*gem*-bisphosphonic acids **136a**–**c** in 73 to 88% yield (Figure 63). The compounds obtained were evaluated as effective inhibitors of *Mycobacterium tuberculosis* glutamine synthetase [29].

In a similar way, the Michael-type addition of alkyl amines to tetraethyl vinyl *gem*-bisphosphonate **134** produced the *N*-alkyl β-amino-*gem*-bisphosphonates **137a**–**c** in 82 to 97% yield, which, by treatment with hydrochloric acid at reflux, gave the *N*-alkyl β-amino-*gem*-bisphosphonic acids **138a**–**c** in 53 to 81% yield (Figure 64). The activity against *Toxoplasma gondii* proliferation at sub-micromolar levels of the obtained compounds was tested [53].

Grigor’ev et al. [54] carried out the addition of 2,2,6,6-tetramethylpiperidine **139** to tetraethyl vinyl *gem*-bisphosphonate **134** to obtain *N*-alkyl β-amino-*gem*-bisphosphonate **140** in 85% yield, and antitumor and cytotoxic activity was tested (Figure 65).

The Michael-type addition of alkyl amines to tetraethyl vinyl *gem*-bisphosphonate **134** produced the *N*-alkyl β-amino-*gem*-bisphosphonates **141a**–**h**, which, by treatment with TMSBr in dichloromethane, gave the *N*-alkyl β-amino-*gem*-bisphosphonic acids **142a**–**h** (Figure 66). The activity against *Trypanosoma cruzzi*, the etiologic agent of American trypanosomiasis (Chagas’ disease), and against tachyzoites of *Taxoplasma gondii* was evaluated for the obtained compounds [23].

Berlicki et al. [31] carried out the additional reaction of aromatic amines to tetraethyl vinyl *gem*-bisphosphonate **134** at room temperature followed by treatment with TMSBr and the subsequent addition of MeOH, obtaining the *N*-aryl β-amino-*gem*-bisphosphonic acids **143a**–**g** in 3 to 60% yield, which are effective inhibitors of *Arabidopsis thaliana* δ^1^-pyrroline-5-carboxylate reductase and *Mycobacterium tuberculosis* glutamine synthetase (Figure 67).

After optimizing the solvent and temperature, Lv et al. [120] carried out the addition of various aromatic amines to tetraethyl vinyl *gem*-bisphosphonate **134** in CHCl_3_ at 60 °C, obtaining the corresponding *N*-aryl β-amino-*gem*-bisphosphonates **144a**–**j** in 46 to 91% yield (Figure 68).

Strukul et al. [55] carried out the addition reaction of amines to tetraethyl vinyl *gem*-bisphosphonate **134** in the presence of Et_3_N in chloroform at room temperature, obtaining the *N*-substituted β-amino-*gem*-bisphosphonates **144a** and **145a,b** in quantitative yield, which, by treatment with TMSBr followed by the addition of H_2_O, produced the *N*-substituted β-amino-*gem*-bisphosphonic acids **146a**–**c** in quantitative yield, which are effective inhibitors of *Arabidopsis thaliana* δ^1^-pyrroline-5-carboxylate reductase and *Mycobacterium tuberculosis* glutamine synthetase (Figure 69).

The addition reactional of amino acetaldehyde dimethyl acetal **147** with tetraethyl vinyl *gem*-bisphosphonate **134** in dichloromethane produced the *N*-substituted β-amino-*gem*-bisphosphonate **148** in 99% yield, while the amino acetaldehyde dimethyl acetal **147** reacted with tetrakis(trimethylsilyl) vinyl *gem*-bisphosphonate **149** in acetonitrile at 20 °C followed by treatment with H_2_O afforded the *N*-substituted β-amino-*gem*-bisphosphonic acid **150** in 63% yield (Figure 70) [121].

The reaction of tetraethyl vinyl *gem*-bisphosphonate **134** with propargylamine in chloroform at 65 °C afforded the β-amino-*gem*-bisphosphonates **151**, which without further purification was reacted under 1,3-dipolar click cycloaddition with the corresponding azides, sodium ascorbate (NaVc) and CuSO_4_·5H_2_O in *t*-BuOH:H_2_O mixture assisted by ultrasound irradiation, obtaining the 1,2,3-triazole-amino-bisphosphonates derivatives **152a**–**h** in 64 to 72% yield (Figure 71) [122].

Xu et al. [123] carried out the reaction of tetraethyl vinyl *gem*-bisphosphonate **134** and sodium azide in EtOH/H_2_O, obtaining the corresponding azide intermediate **153** in 60% yield, which, by hydrogenation using Pd/C in anhydrous MeOH, produced the β-amino-*gem*-bisphosphonate **154**, that without further purification was used in the synthesis of melphalan—MMP-2-linkage—bisphosphonic acid **155**, evaluated as an anticancer prodrug. Additionally, the reaction of tetraethyl vinyl *gem*-bisphosphonate **134** with trimethylsilyl azide in the presence of Pd(OAc)_2_ in toluene at 90 °C, also gives the β-amino-*gem*-bisphosphonate **154** in 90% yield in only one step (Figure 72) [124].

Chen et al. [125] developed a practical and efficient one-pot strategy for the synthesis of *N*-attached 1,2,3-triazole-containing bisphosphonates, integrating two key transformations into a single operation. Thus, the Michael addition of sodium azide to tetraethyl vinyl *gem*-bisphosphonate **134** in AcOH/H_2_O under mild conditions and ultrasonication gave the corresponding azide derivative **153**, which, by a copper-catalyzed 1,3-dipolar, was reacted with terminal alkynes in the presence of CuSO_4_·5H_2_O and sodium ascorbate (NaVc) to obtain the *N*-triazole-functionalized *gem*-bisphosphonates **156a**–**m** in 73 to 84% yield (Figure 73).

The reaction of the *gem*-bisphosphonate **157** with NaH followed by the addition of *N*-(bromomethyl)phthalimide gave the alkylated product **158** in only low yield. Best results were obtained when the conjugate addition of phthalimide to tetraethyl vinyl diphosphonate **134** in dimethylformamide was carried out, obtaining the *N*-substituted β-amino-*gem*-bisphosphonate **159** in 73% yield, which, by alkylation with propargyl bromide using NaH as a base, produced the acetylene derivative **160** in 80% yield. The 1,3-dipolar click cycloaddition with geranyl azide afforded the corresponding triazole derivative **158** in 40% yield, used as a key intermediate for the synthesis of α-modified triazole bisphosphonic salts **161**, which were evaluated for their activity as GGDPS inhibitors in both enzyme and cell-based assays (Figure 74) [126].

The additional reaction of thiourea arylamines derived from dehydroabietic acid **162** to tetraethyl vinyl *gem*-bisphosphonate **134** in the presence of dimethylaminopyridine (DMAP) in dichloromethane at room temperature produced the corresponding *N*-substituted β-amino-*gem*-bisphosphonates **163a**–**k** in 73 to 93% yield. These compounds exhibited potent antitumor activity against the SK-OV-3, BEL-7404, A549, HCT-116 and NCI-H460 tumor cell lines in vitro; especially, the β-amino-*gem*-bisphosphonate **163d** exhibited the best anticancer activity against the SK-OV-3 cell line (Figure 75) [127].

The Aza-Michael addition of *O*-benzylhydroxylamine to tetraethyl or tetrabenzyl vinyl *gem*-bisphosphonates **134** and **164** in CH_2_Cl_2_ or Et_3_N in CH_2_Cl_2_ at room temperature produced the corresponding β-amino-*gem*-bisphosphonate **165a**,**b** fosmidomycin analogs in 40% yield, which were transformed into amides **166a**,**b**. The pro-herbicide activity of these compounds was evaluated on model plants, having considerable herbicidal activity (Figure 76) [56].

Rodriguez et al. [57], through molecular design, found that compounds **168** and **171** could act against *Trypanosoma cruzi* (amastigotes), *Toxoplasma gondii* (tachyzoites), T*c*FPPS and T*g*FPPS. With these results, they carried out the additional reaction of 2-methylallylamine to tetraethyl vinyl *gem*-bisphosphonate **134** in methylene chloride at room temperature, obtaining the β-amino-*gem*-bisphosphonate **167** in 91% yield, which, by reaction with TMSBr in dichloromethane followed by treatment with methanol, produced the β-amino-*gem*-bisphosphonic acid **168** in 83% yield. Additionally, the Aza-Michael addition of (*E*)-2,6-dimethylhepta-2,5-dien-1-amine to tetraethyl vinyl diphosphonate **134** produced the β-amino-*gem*-bisphosphonate **169** in quantitative yield. However, after several attempts of hydrolysis of phosphonic esters, it was not possible to obtain β-amino-*gem*-bisphosphonic acid **170**, therefore β-amino-*gem*-bisphosphonate **169** was treated with acetic anhydride to give acetylated β-amino-*gem*-bisphosphonate **171** in 63% yield (Figure 77). Compound **168** was inactive against *Trypanosoma cruzi* and *Toxoplasma gondii* cells but exhibited a marginal activity against the target enzymes T*c*FPPS and T*g*FPPS.

The reaction of tetraethyl vinyl *gem*-bisphosphonate **134** with *NH*-3,5-bis(arylidene)piperid-4-one **172** and triethylamine in dichloromethane at room temperature gave the corresponding *N*-substituted β-amino-*gem*-bisphosphonates **173a**–**g** in 72 to 99% yield. The reaction of the *N*-substituted β-amino-*gem*-bisphosphonates **173b**–**d** with TMSBr in CHCl_3_ at room temperature followed by treatment with methanol gave the *N*-substituted β-amino-*gem*-bisphosphonic acids **174a**–**c** in 56 to 99% yield (Figure 78). The synthesized compounds displayed high inhibitory properties towards Caov3, A549, PC3 and KB 3-1 human carcinoma cell lines, among those, compounds bearing 4-cyano-phenyl **173d** and 3-pyridinyl **173f**, identified as the most active drug candidates possessing fluorescence properties that could be of interest for the visualization of BP skeletal distribution and cellular uptake in bones and other tissues [58].

The Michael-type addition of *N*-Boc diamines to tetra isopropyl vinyl *gem*-bisphosphonate **175** in toluene at reflux gave the β-amino-*gem*-bisphosphonates **176a**,**b** in 78 and 81% yield, respectively, which, by hydrolysis with 4 M HCl, produced the β-amino-*gem*-bisphosphonic acids **177a**,**b** in 71 and 76% yield, respectively. In a similar way, the reaction of tetraamine with tetra isopropyl vinyl *gem*-bisphosphonate **175** gave the β-amino-*gem*-bisphosphonate **178** in 83% yield, which, by hydrolysis with 4 M HCl, afforded the β-amino-*gem*-bisphosphonic acid **179** in 70% yield (Figure 79). The affinity of these compounds to hydroxyapatite was also determined by using the ^99m^Tc procedure [59].

The Michael-type addition of 2-amino-6-chlorobenzothiazole **180** to tetraethyl vinyl *gem*-bisphosphonate **134** in CHCl_3_ at 40 °C produced the *N*-substituted β-amino-*gem*-bisphosphonate **181** in 65% yield, which, by hydrolysis with 3 N HCl, afforded the *N*-substituted β-amino-*gem*-bisphosphonic acid **182** in 96% yield (Figure 80). Compound **182** was evaluated in an enzyme inhibition assay against MMP-2, MMP-8, MMP-9 and MMP-13 [41].

Tsantrizos et al. [128] carried out the reaction of 1-aminoimidazoles **183a**,**b** with tetraethyl vinyl *gem*-bisphosphonate **134** in a CH_2_Cl_2_/DMF mixture at room temperature to obtain the corresponding β-amino-*gem*-bisphosphonates **184a**,**b** in 56% and 60% yield, respectively, which, by treatment with TMSBr in dichloromethane followed by the addition of methanol, produced the *N*-substituted α-amino-*gem*-bisphosphonic acids **185a**,**b** in 80% and 94% yield, respectively (Figure 81). They also explored the interactions of these compounds with both the active site and an allosteric pocket of human farnesyl pyrophosphate synthase (hFPPS) to evaluate their potential utility in the treatment of cancer and neurodegenerative diseases.

Vagapova et al. [129,130] carried out the reaction of either tetraethyl vinyl *gem*-bisphosphonate **134** or tetrakis(trimethylsilyl) vinyl *gem*-bisphosphonate **149** with amine **186** in dioxane at 100 °C, obtaining the *N*-substituted α-amino-*gem*-bisphosphonates **187a**–**d** in 52 to 96% yield and *N*-substituted α-amino-*gem*-bisphosphonic acids **188a**–**c** in 42 to 61% yield (Figure 82). These compounds showed a potential antiresorptive and antiproliferative activity.

The reaction of tetraethyl vinyl *gem*-bisphosphonate **134** with Rhodamine B derivatives **189a**,**b** in the presence of DMAP in CH_2_Cl_2_ at room temperature afforded the β-amino-*gem*-bisphosphonates **190a**,**b** both in 7% yield (Figure 83). The fluorescent probes of these compounds were evaluated for theranostic applications, showing promising properties such as rapid response, optical stability, hydroxyapatite sensitivity and low toxicity [131].

The reaction of tetrakis(trimethylsilyl) vinyl *gem*-bisphosphonate **149** with several alkyl or aryl amines and α-amino acids was carried out in dichloromethane or acetonitrile or toluene at room temperature or 80 °C depending on the amine used, followed by treatment with methanol, giving the *N*-substituted β-amino-*gem*-bisphosphonic acids **146c**, and **191a**–**m** in 57 to 93% yield (Figure 84) [60].

The additional reaction of *S*-trityl cysteamine to tetraethyl vinyl *gem*-bisphosphonate **134** in CH_2_Cl_2_ at room temperature produced tetraethyl (2-((2-(tritylthio)ethyl)-amino)ethane-1,1-diyl) bis(phosphonate) **192** in 95% yield, which, by reaction with triethylsilane and trifluoroacetic acid at room temperature, gave the *N*-substituted β-amino-*gem*-bisphosphonate **193** in 93% yield (Figure 85). This compound is a key intermediate for the synthesis of potential anti-resorption bone drugs [132].

Tessmar et al. [133] carried out the reaction of 3,5-diaminobenzoic acid **194** with tetraethyl vinyl *gem*-bisphosphonate **134** in THF at 60 °C, obtaining 3,5-di(tetraethylamino-2,2-bisphosphonate)benzoic acid **195** in 85% yield, which, by treatment with TMSBr followed by the addition of methanol, produced β-amino-*gem*-bisphosphonic acid **196**, which was incorporated into PEG-based polymer **197**, enabling the formation of stable gold nanoparticle (GNP) coatings with a strong affinity for hydroxyapatite, making them promising candidates for bone-targeted delivery systems (Figure 86).

The reaction of 4,9-dioxa-1,12-dodecanediamine **198** and tetraethyl vinyl *gem*-bisphosphonate **134** in dry DMF at room temperature gave *gem*-bi(bisphosphonate) derivative **199** in 80% yield. This compound was transformed into the cross-linker **200** and incorporated into hydrogels to evaluate their properties (Figure 87) [134,135].

### 3.2. Addition of gem-Bisphosphonates to Imines

The addition of carbanion derived from *gem*-bisphosphonates to imines of Schiff bases is another method for the synthesis of β-amino-*gem*-bisphosphonates. For example, Abdou et al. [136] carried out the reaction of amine **201** with several aldehydes to obtain the corresponding imine; in situ, these were reacted with tetraethyl *gem*-bisphosphonate and LiOH in DMF at reflux, affording the tetraethyl *gem*-bi(bisphophonates) **202a**–**j** in 66 to 78% yield, which, by treatment with TMSBr followed by reaction with KOH in methanol, gave the *N*-substituted β-bis(amino-*gem*-bisphosphonic) acids as potassium salt **203a**–**j** in ~68% yield (Figure 88). Cytotoxic properties were evaluated for the compounds obtained against five malignant melanoma cell lines that originated from different categories of malignant melanoma primary stage (I/II), histologically advanced stage (III/IV) and metastasized malignancy.

The reaction of Schiff bases derived from 4-(4-methylphenyl)-2,3-benzoxazin-1-one **204a** or 3-phenyl-2,4-benzoxazin-1-one **204b** and aromatic aldehydes, with tetraethyl *gem*-bisphosphonate and LiH in DMF at room temperature, gave the *N*-substituted β-amino-*gem*-bisphosphonates **205a**–**d** in 59 to 72% yield, (Figure 89). The cytogenetic activity in normal human lymphocyte cultures of the obtained compounds was evaluated [61].

The Schiff bases **206a**,**b** derived from 2-aminobenzenethiol and aromatic aldehydes were reacted with tetraethyl *gem*-bisphosphonate and LiH in DMF at reflux, obtaining the corresponding *gem*-bisphosphonates **207a**,**b** in 68 and 62% yield, respectively, which, by treatment with TMSBr followed by reaction with NaOH in methanol, gave the *N*-substituted β-amino-*gem*-bisphosphonic acids as sodium salt **208a**,**b** in 68 and 62% yield, respectively (Figure 90). The compounds obtained were evaluated in a mouse model of antigen-induced arthritis and the delayed-type hypersensitivity garanuloma reaction (DTH-GRA) for chronic inflammation [137].

## 4. Synthesis of γ-Amino-*gem*-Bisphosphonate Derivatives

For the synthesis of γ-amino-*gem*-bisphosphonates, this review described only the addition of carbanions derived from amino acids to vinyl *gem*-bisphosphonates (Figure 91).

### Synthesis from Vinyl gem-Bisphosphonates

Wang et al. [62] carried out a detailed study of the catalytic asymmetric Michael addition for the synthesis of optically active γ-amino-*gem*-bisphosphonates containing two adjacent stereogenic centers. In this context, the reaction of vinyl *gem*-bisphosphonates **209a**–**r** with *N*-benzylidene glycine methyl ester **210** in the presence of catalytic amounts of (*R*)-**211**, CuBF_4_ and K_2_CO_3_ in dichloromethane at −20 °C followed by treatment with *p*-toluenesulfonic acid at room temperature afforded the γ-amino-*gem*-bisphosphonates **212a**–**r** in 62 to 95% yield and 89 to 99% enantiomeric excess (Figure 92).

In a similar way, Fukuzawa et al. [63] carried out the reaction of vinyl *gem*-bisphosphonates **209** with methyl *N*-diphenyleneglycinate **213** in the presence of catalytic amounts of ThioClickFerrophos (TCF), AgOAc and Cs_2_CO_3_ in THF at −20 °C, obtaining the γ-amino-*gem*-bisphosphonates **214a**–**l** in 64 to 99% yield and 90 to 98% enantiomeric excess (Figure 93).

On the other hand, the reaction of vinyl *gem*-bisphosphonate **134** with α-substituted azlactones **215a**–**f** and Et_3_N in dichloromethane at room temperature afforded the compounds **216a**–**f** in 44 to 88% yield and 5:1 to >20:1 regioisomeric ratio, which, by treatment with chlorotrimethylsilane in MeOH:CH_2_Cl_2_ at room temperature, gave the corresponding γ-amino-*gem*-bisphosphonic acids **217a**–**f** in 43 to 58% yield (Figure 94) [138]. In this work, the stereoselective version with α-substituted azlactone **215** and different chiral Brønsted bases was evaluated.

## 5. Synthesis of Heterocyclic α-Amino-*gem*-Bisphosphonate Derivatives

### 5.1. Phosphonylation of Lactams

The bisphosphonylation of lactams with dialkyl or trialkyl phosphites using activating agents such as PCl_3_, P(O)Cl_3_ and Tf_2_O is an efficient and general method for the synthesis of heterocyclic α-amino-*gem*-bisphosphonate derivatives. For example, Kafarski et al. [139] carried out the reaction of γ-, δ- and ε-lactams **218a**–**f** with triethyl phosphite and phosphoryl chloride at 0 °C, obtaining the heterocyclic α-amino-*gem*-bisphosphonates **219a**–**f** in 28 to 65% yield. Additionally, the reaction of benzoannulated lactams **218g**,**h** under identical conditions gave the corresponding heterocyclic α-amino-*gem*-bisphosphonates **219g**,**h** in 17 to 30% yield, accompanied by benzo-fused monophosphonates **220a**,**b** in 30 and 20% yield, respectively (Figure 95).

We developed a versatile methodology for the synthesis of heterocyclic α-amino-*gem*-bisphosphonates from lactams using two distinct approaches. In the first strategy (Method A), the *N*-benzyl piperidin-2-one **221a**, morpholin-3-one **221b** and thiomorpholin-3-one **221c** were reacted with triethyl phosphite and phosphoryl chloride, affording the corresponding heterocyclic α-amino-*gem*-bisphosphonates **222a**–**c** in 20 to 87% yield. Treatment of **222b**,**c** with TMSBr followed by the addition of methanol gave the heterocyclic α-amino-*gem*-bisphosphonic acids **223b**,**c** in 60% yield. Additionally, the reaction *N*-benzylated thiomorpholin-3-one **221d** with triethyl phosphite Tf_2_O DTBMP (Method B) gave the heterocyclic α-amino-*gem*-bisphosphonate **222d** in 70% yield. These heterocyclic α-amino-*gem*-bisphosphonates constitute valuable building blocks for medicinal chemistry, particularly due to their potential bone-targeting properties and structural relevance to bioactive bisphosphonates (Figure 96) [140].

Wang et al. [107] reported a mild and efficient bisphosphonylation protocol of lactams based on the activation with Tf_2_O. Thus, the reaction of *N*-benzyl lactams **221** with Tf_2_O, diethyl phosphite and DTBMP at 0 °C produced the desired heterocyclic α-amino-*gem*-bisphosphonates **222** in 72 to 93% yield. These reaction conditions proceed with excellent chemoselectivity and tolerate the presence of *tert*-butoxycarbonyl groups (Figure 97).

### 5.2. Miscellaneous

Kolodiazhnyi et al. [141] found that, during the treatment of ethyl isoindolin-1-one-3-yl-phosphonate **224** in acidic conditions, this compound undergoes a retro-Michael reaction, releasing diethyl phosphite and generating imine **225**, which is susceptible to autoxidation and diphosphonylation with diethyl phosphite to give the heterocyclic α-amino-*gem*-bisphosphonate **227**, an analog of benzo[*c*]pyroglutamic acid via the carbon-centered radical **226** (Figure 98).

The reaction of *o*-phthaloyl chloride **228** with the sodium salt of diethyl phosphite in diethyl ether at 0 °C afforded 3,3-bis(diethylphosphono)-1(3*H*)-isobenzofuranone **229** in 40% yield, which, by reaction with benzylamine and triethylamine in toluene at reflux, produced tetraethyl (2-benzyl-3-oxoisoindolyl-1,1-diyl)-bisphosphonate **230** in 41% yield. The reaction of **230** with HCl at reflux gave 2-benzyl-3-oxoisoindolin-1,1-dyil-bisphosphonic acid **231** in 87% yield (Figure 99) [142,143].

On the other hand, the reaction of tetraethyl methylenebisphosphonate with 2*H*-3,1-benzothiazine-2,4(1*H*)-dithiones **232a**,**b** and sodium ethoxide in ethanol at reflux followed by treatment with HCl at −5 °C gave the 3-thioxoindoline-2,2-diyldiphosphonates **234a**,**b** in 41 and 40% yield, respectively, via the intermediates **233a**,**b** (Figure 100). Similar results were obtained in the reaction of 1*H*-benzo[*d*][1,3]-oxazin-2,4-diones **235a**,**b** with tetraethyl methylenebisphosphonate, affording the tetraethyl 3-oxoindoline-2,2-diyldiphosphonates **236a**,**b** in 58 and 54% yield, respectively. These compounds exhibited remarkable antitumor activity against four tested carcinoma cell lines and showed also significant to moderate anti-inflammatory activity capable of inhibiting polyarthritis [144], and their cytogenetic activity in normal human lymphocyte cultures was also evaluated [61].

Scarso et al. [145] reported the synthesis of a large series of substituted pyrrolidine containing bisphosphonates through 1,3-dipolar cycloaddition of azomethine ylides to vinyl *gem*-bisphosphonates as dipolarophiles. Thus, the 1,3-dipolar cycloaddition of azomethine ylides **237** derived from the condensation/decarboxylation of aldehydes with the corresponding α-amino acids or sarcosine, and the tetraethyl vinyl *gem*-bisphosphonates **209**, afforded the (pyrrolidine-3,3-diyl)bis(phosphonate) esters **238a**–**n** in 7 to 68% yields. This strategy represents a straightforward and modular approach to access structurally diverse *N*-heterocyclic *gem*-bisphosphonates, with potential application in bone resorption diseases such as osteoporosis (Figure 101).

Finally, the 1,3-dipolar cycloaddition of azomethine ylides derived from the condensation/decarboxylation of aldehydes with diethyl 2-aminomalonate **239** and tetraethyl vinyl *gem*-bisphosphonate **134** afforded the diethyl 4,4-bis(diethoxyphosphoryl)-5-pentylpyrrolidine-2,2-dicarboxylates **240a**–**c** in 41 to 68% yield (Figure 102) [146].

## 6. Concluding Remarks

In this review, we have covered the last fifteen years in the development of new synthetic methodologies for the synthesis of acyclic and heterocyclic amino-*gem*-bisphosphonic acids and derivatives; the biological activities have also been discussed.

All these methodologies give the synthetic organic chemist the opportunity to select the most appropriate way to obtain the desired acyclic and heterocyclic amino-*gem*-bisphosphonic acids and derivatives.

We propose that continued efforts should be made to search for and improve synthetic procedures for the preparation of new amino-*gem*-bisphosphonates, and that the exploration of new chemical and biological applications of these interesting compounds will be a very rewarding task for researchers in this area and necessary for the coming years.

## Data Availability

No new data were created or analyzed in this study. Data sharing is not applicable to this article.

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
