# Peer review of "Synthesis of Amino-gem-Bisphosphonate Derivatives and Their Application as Synthons for the Preparation of Biorelevant Compounds"

_pharmaceuticals, 2025, doi:10.3390/ph18071063_

Round 1
Reviewer 1 Report
Comments and Suggestions for Authors
See attached file

The English should be polished.
Author Response
Enclosed you will find our revised manuscript ID: pharmaceuticals-3737815 “Synthesis of amino-gem-bisphosphonate derivatives and their application as synthons for the preparation of biorelevant compounds”
We would like to sincerely thank the professors (referee 1), who reviewed our manuscript; their comments have been of great support to us.
The revision of our manuscript was conducted following all comments of the referees and the corrections were highlighted in yellow color. The spot-to-spot response is following.
Reviewer 1.
This reviewer believes that this review article may be accepted for publication in Pharmaceuticals after revision.
The following points should be addressed:
- Line 14. It is said “because present important biological properties and are also used as valuable key...”, it would be better to say “due to their important biological properties and their value as key...”
Response. Changes done. The indicate sentences and words were modified.
- Line 59–60. The verb is missing in the sentence “Additionally, gem-bisphosphonic acids and gem-bisphosphonates 59 analogues of pyrophosphoric acid and pyrophosphate, respectively...” It should be better to say “Additionally, gem-bisphosphonic acids and gem-bisphosphonates 59 are analogues of pyrophosphoric acid and pyrophosphate, respectively...”
Response. Changes done. The indicate sentences and words were modified.
- Sometimes drug names are capitalized, such as Incadronate in line 71. However, in other cases, the authors write drug names in lowercase, such as zoledronate in line 181. Please unify the formatting throughout the manuscript.
Response. The change has been made. The formatting of drug names has been unified throughout the manuscript.
- Line 71. Substitute “act” by “acts”.
Response. The change has been made. “Act” has been replaced by “acts” in line 71.
- Figure 2. Substitute “PB-Btz” by “BP-Btz”. There is an extra “a” in the name of compound 7
Response. The change has been made. “PB-Btz” has been replaced by “BP-Btz” in Figure 2, and the extra “a” in the name of compound 7 has been removed.
- Line 93 and 99. Substitute orthoformates instead of orthoformate.
Response. The change has been made. “Orthoformate” has been replaced by “orthoformates” in lines 93 and 99.
- There are some typos in the manuscript. For example: line 90 properties instead of proprieties, line 106 moderate instead of moderated, line 213 and 223 heterocyclic instead of heterocycle. Please check the manuscript again to see if there are any other typos.
Response. The changes have been made. The indicated typos have been corrected, and the entire manuscript has been carefully reviewed for any additional typographical errors.
- Line 123. Yields of compounds 10a–v from 32 to 89 (10j has the lower yield).
Response. The change has been made. The yields of compounds 10a–v are now reported as 32 to 89%.
- Line 126. Yields of compounds 11a–v from 21 to 98 (11u has the highest yield).
Response. The change has been made. The yields of compounds 11a–v are now reported as 21 to 98%.
- Line 140. Compounds 14 are referred to as bisphosphonates, however they should be better referred to as bis-H-phosphinates.
Response. The change has been made. Compounds 14 are now referred to as bisphosphinates.
- Line 147. Compounds 11 are not bisphosphonates but bisphosphonic acids.
Response. The change has been made. Compounds 11 are now referred to as bisphosphonic acids.
- Line 156. Substitute “its” by “their”.
Response. The change has been made. “Its” has been replaced by “their” in line 156.
- Line 170. Substitute “bisphosphonate 17 was use also...” by “bisphosphonate 17 was also used...”
Response. The change has been made. “Bisphosphonate 17 was use also...” has been replaced by “bisphosphonate 17 was also used...” in line 170.
- Sch. 9. Compound 17a (on the middle of scheme) should be referred to as compound 17, and compound 17f should be corrected to 17e.
Response. The change has been made. Compound 17a is now referred to as compound 17, and compound 17f has been corrected to 17e in Scheme 9.
- Line 186. There is an extra opening parenthesis.
- Response. The change has been made. The extra opening parenthesis in line 186 has been removed.
- Line 210. The final period is missing.
Response. The change has been made. The final period has been added in line 210.
- Scheme 13: Should the NH in structure 24 be represented as NR, where R = THP, Me? The definition of Ar in structure 27a is missing.
Response. The clarification has been made. THP and Me belong to molecule 25, and the definition of Ar in the structures has been added.
- Line 232 and 234: Substitute “33a-n” and 34a-n by “33a-q and 34a-q". The same in Sch. 16.
Response. The correction has been made. “33a-n” and “34a-n” have been replaced by “33a-q” and “34a-q” in lines 232 and 234, as well as in the corresponding scheme.
- Sch.16: The yields of compounds 33a-q are missing.
Response. The yields of compounds 33a–q have been added.
- Line 272: "...with several aryl and heteroaryl amines, diethyl phosphite..." should be replaced with "...with several aryl and heteroaryl amines, and diethyl phosphite...".
Response. The correction has been made. The phrase has been revised to “…with several aryl and heteroaryl amines, and diethyl phosphite…” now in line 318.
- Line 302: Replace "10a,j,l” with “10h,j,l”.
Response. The correction has been made. “10a,j,l” has been replaced with “10h, 10j, 10l, 58b, 58i, 58k,l, and 60a-m”, and this change now appears in line 375.
- Line 309: Replace "10h,j,l" with "10b,e,p".
Response. The correction has been made. “10h,j,l” has been replaced with “10b, 10e, 10p, 29a and 61a-f”, and this change now appears in line 382.
- Line 310: It is said "antioxidant properties in nitric oxide and antimicrobial activity..." it would be better to say "antioxidant properties in nitric oxide method inhibitory potency and antimicrobial activity...".
Response. The text has been revised as suggested. The phrase now reads “antioxidant properties in nitric oxide method inhibitory potency and antimicrobial activity…” in line 383.
- Line 320: Replace "49a" with "49b".
Response. The modifications have been made. They can now be seen in line 394 as: 10b–f, 29a, 58, 58o,p, and 62a–g.
- Sch. 25: Compounds 52 does not correspond with compounds 52 in sch. 24.
Response. The change has been made.
- Text of sch. 26: There is some confusion regarding the conditions for methods A and B. According to the description under the scheme, method A involves conventional heating, while method B uses ultrasonication. However, these conditions are not clearly defined in the text.
Response. This issue has been addressed. The conditions for methods A (conventional heating) and B (ultrasonication) are now clearly defined in the text. Now scheme 28.
- Line 332: Reference 68 does not correspond to the catalyzed three-component reaction reported in sch. 27.
Response. The observation has been addressed.
- Sch.27: Compound 52j is not the same as compound 52j in sch.26.
Response. The observation has been addressed. Now scheme 28.
- Line 348: Replace "catalyst" with "catalysts".
Response. The correction has been made. “Catalyst” has been replaced with “catalysts” now in line 327.
- Line 349: I suggest replace "better catalyst" with "the most effective catalyst".
Response. The suggested change has been implemented. “Better catalyst” has been replaced with “the most effective catalyst” now in line 328.
- Line 353: "53a-f" instead of "53af".
Response. The observation has been addressed. Now in line 332.
- Line 356: "One compound..." should be changed to "One of the compounds tested..." or directly "Compound 46k...".
Response. The observation has been addressed. Now in line 336.
- Line 365: Reference 70 does not correspond to the catalyzed three-component reaction reported in sch. 29 using amino acid derivatives.
Response. The observation has been addressed.
- Sch.29: "55d" instead of "545".
Response. The observation has been addressed. Now scheme 34.
- In Section 2.1, the authors report several methods for the three-component reaction of orthoformates, amines, and dialkyl phosphites. Some of these are uncatalyzed, while others involve the use of catalysts. The authors begin by reporting the uncatalyzed methods (Schemes 2 to 19), followed by the catalyzed ones (Schemes 20 to 29). Next, they describe several uncatalyzed methods involving microwave irradiation (Schemes 30 to 34). I suggest that the authors move Schemes 30 to 34 to be grouped with the other uncatalyzed methods, before introducing the catalyzed versions of this three-component reaction.
Response. The methodologies have been appropriately grouped as suggested.
- Sch. 30: R' in compound 56a is n-Bu instead of Bu. There is a missing spaces in 56f.
Response. The necessary corrections have been made. R' in compound 56a is now shown as n-Bu, and the missing spaces in 56f have been added.
- Line 399: were tested for in vitro. Their anti-inflammatory activity showed moderate inhibition...", it would be better to say were tested for in vitro antiinflamatory activity, showing moderate inhibition...".
Response. The text has been revised as suggested. It now reads: “were tested for in vitro anti-inflammatory activity, showing moderate inhibition...”
- Sch.33: Substituents in bisphosphonates are not correct.
Response. The observation has been addressed.
- Sch.36: Compound 73 is the same as 17.
Response. The correction has been made. Compound 73 is now indicated as the same as 17 in Scheme 36.
- Sch. 37: There are some missing spaces in 77b and 11b.
Response. The observation has been addressed.
- Line 458: Which Lewis acid is used in this reaction? ZnCl2 in the text but Znl2 in the sch.38.
Response. The correction has been made. ZnIâ‚‚ is now indicated as the Lewis acid used in both the text and Scheme 38.
- Line 487: Which conditions are used in this reaction? R.t. or 80 °C in the text but r.t. to 80 °C in sch.
Response. The clarification has been made. The conditions are r.t. or 80 °C, and this is now specified in the scheme.
- Line 491: One of the key characteristic of multicomponent reactions (MCRs) is that the final product incorporates most, if not all, of the starting materials. According to the definition I do not think this reaction is a three-component reaction. The same in line 509. Line 518 and 526, in both cases TMSOTf is used as a catalyst, therefore, these are not multicomponent reactions. Line 533-534, this is a stepwise reaction (3 steps) and not a multicomponent reaction.
Response. The manuscript has been revised accordingly. The reactions in lines 491, 509, 518, 526, and 533–534 are no longer described as multicomponent reactions.
- Line 515: The yield of 98 is different in the text and in sch. 45.
Response. The inconsistency has been corrected. The yield is now reported consistently in both the text and Scheme 45.
- Sch. 47: The yields of compounds 106 are missing.
Response. The yields of compounds 106 have been incorporated into Scheme 47.
- Sch. 48: Replace "107a-f and 108a-f” with "107a-n and 108a-n".
Response. The correction has been made. “107a-f and 108a-f” has been replaced with “107a-n and 108a-n” in Scheme 48.
- Line 547: font type inconsistencies: “2.4."
Response. The font type inconsistencies have been corrected in line 547.
- Sch. 50: Delete Br and replace it with R in structure 112k. Compound 112p and 112q have the same substituent R = F.
Response. The observation has been addressed.
- Line 569: An oxygen is missing in the formula NiCl2.6H2.
Response. The observation has been addressed.
- Line 588: Replace "Ti-Mediated” with “Titanium-mediated".
- Sch 55: Nitrile 119 is the same as 111 in sch. 50 and 113 in sch. 51. The same in sch. 56 and the same in sch. 58.
Response. The correction has been made. “Ti-Mediated” has been replaced with “Titanium-mediated” in line 588.
- Line 615: "anti-proliferative effect MCF-7 human breast cancer cells, cells J774E mouse macrophages...", it would be better to say "anti-proliferative effect on MCF-7 human breast cancer cells, J774E mouse macrophages cells...".
Response. The suggested revision has been made. The text now reads: “anti-proliferative effect on MCF-7 human breast cancer cells, J774E mouse macrophages cells…”.
- Sch. 58: Compound 124b and 125b, the letter "b" is missing.
Response. The observation has been addressed.
- Line 621: Yields of compounds 8b, 8j, 8k and 10y, 10z from 50 to 74% (8b has the highest yield).
Response. The yields of compounds 8b, 8j, 8k and 10y, 10z have been included as 50 to 74%.
- Line 623: Change "10y,z” to “11y,z”. Yields of compounds 9b, 9j,k and 11y,z from 69 to 97% (9b has the highest yield).
Response. The change has been made.
- Sch. 59: Nitrile 126 is the same as 111 in sch. 50 and 113 in sch. 51...
Response. The change has been made.
- Sch. 63 and text: Does compound 136a have a yield of 86% or 88%?
Response. The yield of 136a is 88%. This correction has been made in both the text and the scheme.
- Line 674: Is author second name Grigor'ev (Line 674) or Grigor'eva (in the reference section)?
Response. The suggestions have been addressed.
- Line 676: It should be better to mention that oncophos is the bisphosphonate 140. The same in sch.
Response. The suggestions have been addressed.
- Sch. 66: Compounds 141d and 141e have the same substituent, R = n-undecyl. The same for compounds 142d and 142e.
Response. The suggestions have been addressed.
- Line 704: Change "145b,c" to "145a,b".
Response. The suggestions have been addressed.
- Line 755: "...bisphosphonic salts 161, evaluated for activity as GGDPS inhibitors in enzyme and...", it would be better to say "...bisphosphonic salts 161, which were evaluated for their activity as GGDPS inhibitors in both enzyme and...".
Response. The text has been revised as suggested. It now reads: “…bisphosphonic salts 161, which were evaluated for their activity as GGDPS inhibitors in both enzyme and…”.
- Line 776: "Rodriguez et al. by molecular design they found that the compounds...", it would be better to say "Rodriguez et al. through molecular design found that compounds...".
Response. The text has been revised as suggested. It now reads: “Rodriguez et al. through molecular design found that compounds…”.
- Line 787: Change compound "162" to "171".
Response. The suggestions have been addressed.
- Sch. 84: Compound 146c does not correspond to compound 146c in sch. 69. Compound 191n is missing.
Response. The suggestions have been addressed.
- Line 887: Do compounds 203a-j have the same chemical yield?
Response. The suggestions have been addressed.
- Sch. 88: Compounds 203a-j and their yields are missing.
Response. The suggestions have been addressed.
- Sch. 90: Yields of compounds 207a, b and 208a,b are different than those reported in the text.
Response. The yields of compounds 207a, b and 208a, b have been updated to ensure consistency between the text and Scheme 90.
- Line 912: "For the synthesis of y-amino-gem-bisphosphonates, in this review only is described...", it would be better to say "For the synthesis of y-amino-gem-bisphosphonates, this review describes only...".
Response. The text has been revised as suggested. It now reads: “For the synthesis of γ-amino-gem-bisphosphonates, this review describes only…”.
- Line 924: Yield 70 for compounds 212a-r should be replaced with 62 (212qb has the lowest yield).
Response. The suggestions have been addressed.
- Sch. 94: The structure of compounds 216a-f are incorrect.
Response. The suggestions have been addressed.
- Sch. 100: Yields under the structure of intermediate 233a, b should be deleted.
Response. The suggestions have been addressed.
- Line 1017: “Finally, the 1,3-dipolar cascade cycloaddition reaction 1,3-dipolar cycloaddition of...", it would be better to say "Finally, 1,3-dipolar cycloaddition of...".
Response. The text has been revised as suggested. It now reads: “Finally, 1,3-dipolar cycloaddition of…”.
- Ref. 15: There are some missing spaces between name initials.
Response. The suggestions have been addressed.
Ref. 57, 58, 61, 68, 70, 73: The author Cirandur is cited differently in some references. Please check for consistency. For instance:
- Ref. 68: TWEEN 20-/H2 O PROMOTED GREEN SYNTHESIS, COMPUTATIONAL AND ANTIBACTERIAL ACTIVITY OF AMINO ACID SUBSTITUTED METHYLENE BISPHOSPHONATES. S. Siva Prasad, S. H. Jayaprakash, Ch. Syamasundar, P. Sreelakshmi, C. Bhuvaneswar, B. VijayaBhaskar, W. Rajendra, S. K. Nayak, and C. Suresh Reddy.
Could you clarify whether "Cirandur” is the first name or the surname?
Response. The suggestions have been addressed.
- Ref. 82: The iso abbreviation of Heteroatom Chemistry is Heteroat. Chem.
Response. The suggestions have been addressed.
- Ref. 89: There is an extra opening parenthesis.
Response. The extra opening parenthesis in reference 89 has been removed.
- Ref. 91: The authors are not properly cited. Wang, A.-E.; ...
Response. The suggestions have been addressed.
- Ref. 143 and 144: Which one is the correct, Zemlianoy or Zemlianoi?
Response. The suggestions have been addressed.
- Concluding Remarks: The Concluding Remarks section would benefit from a brief discussion of a perspective on future directions in this field.
Response. The suggestions have been addressed.
Reviewer 2.
The manuscript is well-structured, extensively referenced and includes detailed reaction schemes. Few suggestions are given to further improve the quality of the manuscript.
- Minor English improvement may help improve readability in some sections.
Response. Thank you for your suggestion. Thanks to the comments from Reviewer 1, the English throughout the manuscript has been substantially improved to enhance readability.
- It would benefit from a table summarizing biological applications and activities of key compounds to enhance clarity.
Response. The suggested table summarizing the biological applications and activities of key compounds has been incorporated into the review.
- Application part of the reaction schemes can be modified to greater extent. Tabular representation or graphical representation may enhance the understanding of the topic.
Response. The suggestions have been addressed.
- Application as synthons for the preparation of biorelevant compounds can be elaborated further so that the objective of the manuscript gets fulfilled.
Response. In this review, a general retrosynthetic analysis of the use of bisphosphonates as synthons has been incorporated into each section. This approach highlights their application in the preparation of biorelevant compounds and helps to fulfill the objective of the manuscript.
Reviewer 3.
The findings from your article indeed are impressive and well-rounded. The information that you have provided on synthetic methodologies for immunogen biophone nates do a good job in providing valuable insights into traditional as well as novel catalytic systems. Attention to detail and thoroughness of providing references have indeed enhanced the quality of this research. An improvement I would recommend would be to consider using a table format to compare outcomes of your research finding that will help structure and bring to attention comparisons without having to read through long procedural descriptions. I did notice minor typographical errors. As something to keep in mind in future publications, it would be advisable to include a few schematic overviews especially of pharmacological relevance especially in the introduction section that could help tell a story and strengthen foundational understanding of your topic for interdisciplinary audiences.
Response. Thank you very much for your thoughtful and constructive feedback as well as your suggestions for further improvement.
In response to your recommendation, a table summarizing the biological activity of key compounds has been incorporated into the review to facilitate direct comparison and improve clarity. We agree that this format helps readers quickly grasp the main findings without having to navigate lengthy procedural sections.
Thank you again for your valuable comments, which have helped to strengthen the quality and presentation of our manuscript.
Reviewer 4.
Mario Ordóñez and Rubén presented their review manuscript titled, “Synthesis of amino-gem-bisphosphonate derivatives and their application as synthons for the preparation of biorelevant compounds”. The authors aimed to investigate the synthetic strategies published in the last fifteen years for the synthesis of acyclic and heterocyclic α-, β- and γ-amino-gem-bisphosphonates and their biological properties. All the schemes, figures are presented well and no technical errors in the contents of this manuscript. Further, the manuscript is organized well and presented all the relevant information pertaining to the topic and certainly useful to readers to update knowledge on new methods and applications of such compounds.
A careful revision is required before publication:-
- Make corrections to typo: Verocells (Scheme 2).
Response. Thank you for your suggestion. Thanks to the comments from Reviewer 1, the English throughout the manuscript has been substantially improved to enhance readability.
It would benefit from a table summarizing biological applications and activities of key compounds to enhance clarity.
- Provide spacing Bochno and Berlicki[38].
Response. The suggestions have been addressed.
- Microorganisms names must be italic as per the standard guidelines (specifically Page 19).
Response. Thank you for your observation. The names of microorganisms have been italicized throughout the manuscript, including on page 19, in accordance with standard guidelines.
- Wherever applicable, authors need to provide recent literature citations.
Response. Thank you for your observation regarding the inclusion of recent literature citations. This review exclusively covers references from the last 15 years (2010–2025), ensuring that all cited literature is up to date and relevant to current advances in the field.
All references must be formatted according to the journal guidelines
Additionally, the title of the cited articles was included in each of the references.

Reviewer 2 Report
Comments and Suggestions for Authors
The manuscript is well-structured, extensively referenced and includes detailed reaction schemes. Few suggestions are given to further improve the quality of the manuscript.
- Minor English improvement may help improve readability in some sections.
- It would benefit from a table summarizing biological applications and activities of key compounds to enhance clarity.
- Application part of the reaction schemes can be modified to greater extent. Tabular representation or graphical representation may enhance the understanding of the topic.
- Application as synthons for the preparation of biorelevant compounds can be elaborated further so that the objective of the manuscript gets fulfilled.
Author Response
Reviewer 2.
The manuscript is well-structured, extensively referenced and includes detailed reaction schemes. Few suggestions are given to further improve the quality of the manuscript.
- Minor English improvement may help improve readability in some sections.
Response. Thank you for your suggestion. Thanks to the comments from Reviewer 1, the English throughout the manuscript has been substantially improved to enhance readability.
- It would benefit from a table summarizing biological applications and activities of key compounds to enhance clarity.
Response. The suggested table summarizing the biological applications and activities of key compounds has been incorporated into the review.
- Application part of the reaction schemes can be modified to greater extent. Tabular representation or graphical representation may enhance the understanding of the topic.
Response. The suggestions have been addressed.
- Application as synthons for the preparation of biorelevant compounds can be elaborated further so that the objective of the manuscript gets fulfilled.
Response. In this review, a general retrosynthetic analysis of the use of bisphosphonates as synthons has been incorporated into each section. This approach highlights their application in the preparation of biorelevant compounds and helps to fulfill the objective of the manuscript.
Reviewer 3 Report
Comments and Suggestions for Authors
The findings from your article indeed are impressive and well-rounded. The information that you have provided on synthetic methodologies for immunogen biophone nates do a good job in providing valuable insights into traditional as well as novel catalytic systems. Attention to detail and thoroughness of providing references have indeed enhanced the quality of this research. An improvement I would recommend would be to consider using a table format to compare outcomes of your research finding that will help structure and bring to attention comparisons without having to read through long procedural descriptions. I did notice minor typographical errors. As something to keep in mind in future publications, it would be advisable to include a few schematic overviews especially of pharmacological relevance especially in the introduction section that could help tell a story and strengthen foundational understanding of your topic for interdisciplinary audiences.
Author Response
Reviewer 3.
The findings from your article indeed are impressive and well-rounded. The information that you have provided on synthetic methodologies for immunogen biophone nates do a good job in providing valuable insights into traditional as well as novel catalytic systems. Attention to detail and thoroughness of providing references have indeed enhanced the quality of this research. An improvement I would recommend would be to consider using a table format to compare outcomes of your research finding that will help structure and bring to attention comparisons without having to read through long procedural descriptions. I did notice minor typographical errors. As something to keep in mind in future publications, it would be advisable to include a few schematic overviews especially of pharmacological relevance especially in the introduction section that could help tell a story and strengthen foundational understanding of your topic for interdisciplinary audiences.
Response. Thank you very much for your thoughtful and constructive feedback as well as your suggestions for further improvement.
In response to your recommendation, a table summarizing the biological activity of key compounds has been incorporated into the review to facilitate direct comparison and improve clarity. We agree that this format helps readers quickly grasp the main findings without having to navigate lengthy procedural sections.
Thank you again for your valuable comments, which have helped to strengthen the quality and presentation of our manuscript.
Reviewer 4 Report
Comments and Suggestions for Authors
Mario Ordóñez and Rubén presented their review manuscript titled, “Synthesis of amino-gem-bisphosphonate derivatives and their application as synthons for the preparation of biorelevant compounds”. The authors aimed to investigate the synthetic strategies published in the last fifteen years for the synthesis of acyclic and heterocyclic α-, β- and γ-amino-gem-bisphosphonates and their biological properties. All the schemes, figures are presented well and no technical errors in the contents of this manuscript. Further, the manuscript is organized well and presented all the relevant information pertaining to the topic and certainly useful to readers to update knowledge on new methods and applications of such compounds.
A careful revision is required before publication:-
- Make corrections to typo: Verocells (Scheme 2).
- Provide spacing Bochno and Berlicki[38]
- Microorganisms names must be italic as per the standard guidelines (specifically Page 19).
- Wherever applicable, authors need to provide recent literature citations.
All references must be formatted according to the journal guidelines
Author Response
Reviewer 4.
Mario Ordóñez and Rubén presented their review manuscript titled, “Synthesis of amino-gem-bisphosphonate derivatives and their application as synthons for the preparation of biorelevant compounds”. The authors aimed to investigate the synthetic strategies published in the last fifteen years for the synthesis of acyclic and heterocyclic α-, β- and γ-amino-gem-bisphosphonates and their biological properties. All the schemes, figures are presented well and no technical errors in the contents of this manuscript. Further, the manuscript is organized well and presented all the relevant information pertaining to the topic and certainly useful to readers to update knowledge on new methods and applications of such compounds.
A careful revision is required before publication:-
- Make corrections to typo: Verocells (Scheme 2).
Response. Thank you for your suggestion. Thanks to the comments from Reviewer 1, the English throughout the manuscript has been substantially improved to enhance readability.
It would benefit from a table summarizing biological applications and activities of key compounds to enhance clarity.
- Provide spacing Bochno and Berlicki[38].
Response. The suggestions have been addressed.
- Microorganisms names must be italic as per the standard guidelines (specifically Page 19).
Response. Thank you for your observation. The names of microorganisms have been italicized throughout the manuscript, including on page 19, in accordance with standard guidelines.
- Wherever applicable, authors need to provide recent literature citations.
Response. Thank you for your observation regarding the inclusion of recent literature citations. This review exclusively covers references from the last 15 years (2010–2025), ensuring that all cited literature is up to date and relevant to current advances in the field.
All references must be formatted according to the journal guidelines
Additionally, the title of the cited articles was included in each of the references.
Round 2
Reviewer 1 Report
Comments and Suggestions for Authors
The manuscript submitted to Pharmaceuticals by Mario Ordoñez and Rubén Argüello-Velasco presents a comprehensive review of the different methods for the synthesis of amino-gem-bisphosphonate derivatives. The authors have thoroughly revised the manuscript and implemented all the requested modifications in the manuscript as per instructions. In its current form, the manuscript is suitable for publication in this journal.